DOI: 10.1038/s41467-018-03118-x | **OPEN**

# Uncovering the balance of forces driving microtubule aster migration in *C. elegans* zygotes

A. De Simone[1,2], A. Spahr[1], C. Busso[1] & P. Gönczy[1]

Microtubule asters must be positioned precisely within cells. How forces generated by molecular motors such as dynein are integrated in space and time to enable such positioning remains unclear. In particular, whereas aster movements depend on the drag caused by cytoplasm viscosity, in vivo drag measurements are lacking, precluding a thorough understanding of the mechanisms governing aster positioning. Here, we investigate this fundamental question during the migration of asters and pronuclei in *C. elegans* zygotes, a process essential for the mixing of parental genomes. Detailed quantification of these movements using the female pronucleus as an in vivo probe establish that the drag coefficient of the male-asters complex is approximately five times that of the female pronucleus. Further analysis of embryos lacking cortical dynein, the connection between asters and male pronucleus, or the male pronucleus altogether, uncovers the balance of dynein-driven forces that accurately position microtubule asters in *C. elegans* zygotes.

[1] Swiss Institute for Experimental Cancer Research (ISREC), School of Life Sciences, Swiss Federal Institute of Technology Lausanne (EPFL), 1015 Lausanne, Switzerland. [2] Present address: Department of Cell Biology, Duke University Medical Center, Durham, NC 27710, USA. Correspondence and requests for materials should be addressed to P.Gön. (email: Pierre.Gonczy@epfl.ch)

The microtubule aster (hereafter: aster) is a radial array of microtubules growing from an organizing center such as the centrosome, which is tightly associated with the nucleus in most animal cells. Aster positioning is critical for proper cell behavior. For example, in newly fertilized eggs, the two asters associated with the sperm pronucleus ensure efficient meeting with the oocyte pronucleus[1]. Moreover, during division of animal cells, positioning of the two asters that organize the mitotic spindle determines cleavage plane placement[2]. Furthermore, correct positioning of the nucleus associated with asters is important for a wealth of fundamental processes, including efficient spindle assembly and cell migration during development[3]. Despite such crucial importance, the mechanics of aster positioning remain incompletely understood.

Forces that position asters can be generated notably by microtubule polymerization pushing against the cell cortex[4–6] or by molecular motors such as the minus-end directed dynein complex[2,7–9]. Dynein can exert this role by pulling on microtubules while being attached at the cell cortex, the nuclear envelope, or cytoplasmic vesicles. Despite considerable knowledge regarding force generation by microtubules and molecular motors such as dynein, how different force components are integrated in space and time in the cellular context to ensure correct aster and nuclear positioning remains poorly understood.

Aster movements depend on the total force applied by microtubules and molecular motors, as well as on how such force is converted to velocity, in other words on the force–velocity relationship. In the cellular milieu, viscosity dominates over inertia, so that aster force–velocity relationships depend on the viscous drag exerted by the cytoplasm[10]. In this hydrodynamic regime, the drag applied in the direction of motion to a translating rigid body is linearly proportional to its velocity and to a characteristic drag coefficient[10,11] ($F=\gamma v$, $v$: velocity; $F$: force; $\gamma$: drag coefficient), which depends on the shape and orientation of the body as well as on the vicinity of cellular boundaries. In the simple case of a sphere moving through a viscous fluid far from any boundary, the drag coefficient is proportional to the sphere radius and to fluid viscosity, as calculated by Stokes law ($\gamma=6\pi R\eta$; $R$: radius, $\eta$: viscosity).

In the case of asters and associated nuclei, the drag coefficient depends on the extent to which fluid flows through microtubules radiating from centrosomes. Two extreme scenarios can be envisaged. First, the asters-nucleus complex could move as a solid object through which fluid cannot pass, so that it can be approximated with a sphere. Depending on the size of the approximating sphere, this approach under or overestimates the actual drag. In the second extreme scenario, the cytoplasm could flow between microtubules, with independent drags acting on each microtubule and on the nucleus. In this case, the total drag acting on the asters-nucleus is the sum of the individual drags[12], but this is an overestimation. Another consideration is that movements of the asters-nucleus complex generate a flow that is confined by cellular boundaries, thus resulting in a backflow that effectively increases the drag coefficient[13]. Given the above considerations, it has been challenging to achieve theoretical estimates of aster-nucleus complex drag coefficients.

This question was nevertheless investigated through a model that derived such estimates for aster positioning in the one-cell Caenorhabditis elegans embryos by simulating cytoplasm hydrodynamics and calculating the resultant frictional forces exerted by the fluid on each microtubule and the nucleus[14]. The authors first predicted the average drag coefficient of a pronucleus moving without microtubules from the embryo posterior to the center to be 3.3 times that calculated by Stokes law, because of cell boundary effects. Second, they predicted that the average drag

coefficient of a pronucleus-asters complex increases monotonically with microtubule number, reaching a value 3.8 times that of a pronucleus alone when each aster comprises 300 astral microtubules, which corresponds to the number of microtubules that has been estimated experimentally[15,16]. Despite this important theoretical advance, an in vivo determination of the drag coefficient of moving asters has not been achieved in any system, thus preventing to thoroughly understand the force balance imparting proper aster and nuclear positioning in the cellular context.

In this work, we measure the drag coefficient of the male pronucleus on its own and in association with the two microtubule asters using the female pronucleus as in vivo probe in one-cell C. elegans embryos. These drag measurements, together with an analysis of pronuclear movements in embryos depleted of select pools of dynein motors, allows us to uncover the force balance coordinating pronuclear and aster positioning.

## Results

**Measuring drags using the female pronucleus as a probe**. The one-cell C. elegans embryo provides a favorable system to investigate drag coefficients in vivo because of the stereotypical long-range movements of asters and pronuclei (Fig. 1a, b, Supplementary Fig. 1, Supplementary Movie 1, Supplementary Data 1)[17,18]. Initially, the female pronucleus is located at the future embryo anterior, the male pronucleus at the future posterior (Fig. 1a, −168 s; Fig. 1c). The two centrosomes and the asters they organize are associated with the male pronucleus, thus forming the male pronucleus-asters complex (hereafter: male-asters complex or MAC). The male-asters complex first migrates slowly toward the anterior, while the female pronucleus migrates slowly toward the posterior ("slow migration"; Fig. 1a, −168 s; Fig. 1c, d). The two pronuclei then accelerate ("fast migration"; Fig. 1a, −54 s; Fig. 1c, d) and meet in the posterior half of the embryo (Fig. 1a, 0 s; Fig. 1c). Thereafter, the two asters and the joined pronuclei move together toward the cell center ("centration", Fig. 1a, 150 s; Fig. 1c, Supplementary Fig. 1).

Pronuclear migration in one-cell C. elegans embryos requires notably microtubules and dynein[19–21]. It has been proposed that dynein motors anchored on the female pronuclear envelope bind microtubules emanating from the centrosomes and thus pull the female pronucleus toward the male pronucleus during the fast migration phase[1,17,19]. If microtubule asters pull the female pronucleus, a reaction force equal in magnitude and opposite in direction must be exerted on the male-asters complex. These two forces will contribute to the velocities of the two bodies as a function of their respective drag coefficients. Therefore, quantifying the relationship between the two velocities should uncover the ratio between these drag coefficients.

To analyze this relationship, two additional types of forces that act on the male-asters complex must be considered. First, a pulling force exerted on microtubules by dynein anchored at the cell cortex by a ternary complex comprising GOA-1/GPA-16, GPR-1/2, and LIN-5[18]. Second, a dynein-dependent centering force that moves the male-asters complex toward the cell center, independently of the ternary complex and dynein at the nuclear envelope[20,22–24].

We calculated the total force balance acting on the male-asters complex and the female pronucleus, both modeled as translating rigid bodies (Fig. 1b—see Methods for detailed description of underlying assumptions). In brief, the total force applied along the A–P axis on the male-asters complex ($F_{MAC}$) and on the female pronucleus ($F_F$) are balanced each by their respective drag force, which scales with translational velocities $v_{MAC}$ and $v_F$, as well as the average drag coefficients $\gamma_{MAC}$ and $\gamma_F$. Thus, the force

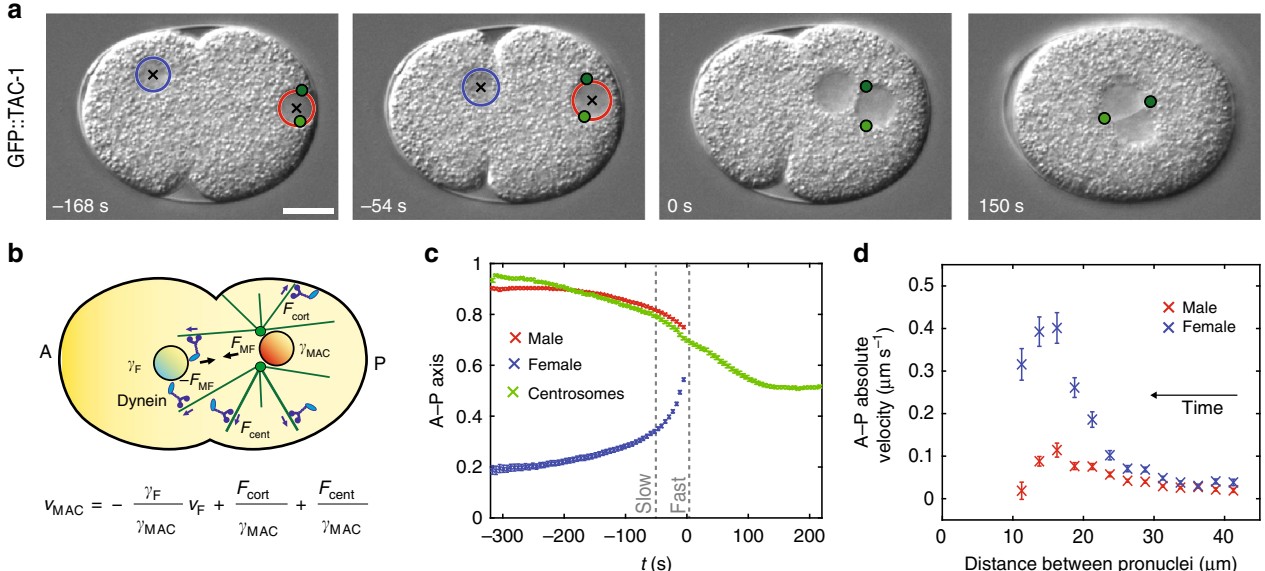

**Fig. 1** 3D time-lapse microscopy of pronuclear migration and centration in the one-cell *C. elegans* embryo. **a** Pronuclei and centrosomes monitored with 3D time-lapse DIC and GFP fluorescence microscopy in embryos expressing GFP::TAC-1. Here and in other figures, representative DIC images are shown; centrosomes (green dots—z-projections), the female pronucleus (blue circle), and the male pronucleus (red circle) are represented; black crosses: pronuclei centers—z-projections. 0 s: pronuclear meeting unless specified otherwise; scale bars: 10 μm. **b** Schematics of forces acting on pronuclei during their migration. Here and in other figures, embryos are schematized with the female and male pronuclei (blue and red disks, respectively), centrosomes (green dots), microtubules (green lines), dynein motors (blue), and dynein anchors (light blue ellipses); arrows represent exerted forces. Anterior (A) and posterior (P) sides are indicated. **c** Average pronuclear and centrosome midpoint positions along the A–P axis as a function of time ($n = 33$, with S.E.M.). Here and thereafter, position on the A–P axis is represented in normalized coordinates (0: anterior; 1: posterior). **d** Absolute average pronuclear velocities as a function of the distance separating them, which decreases over time ($n = 33$, with S.E.M.). Velocities increase while pronuclei approach each other and then diminish when the male-asters complex and the female pronucleus get closer than ~15 μm from one another, probably because of steric hindrance effects. $F_{MF}$ force exerted between the male-asters complex and the female pronucleus, $F_{cort}$ force exerted by cortical dynein, $F_{cent}$ centering force, $\gamma_F$, $\gamma_{MAC}$ drag coefficient of the female pronucleus and male-asters complex, respectively

balance reads

$$F_{MAC} = \gamma_{MAC} v_{MAC} = F_{MF} + F_{cort} + F_{cent},$$
$$F_F = \gamma_F v_F = -F_{MF} \tag{1}$$

where $F_{MF}$ is the nuclear dynein-dependent force exerted between the male-asters complex and the female pronucleus, $F_{cort}$ the cortical force and $F_{cent}$ the centering force. It follows that the velocities of the male and female pronuclei are related as

$$v_{MAC} = -\frac{\gamma_F}{\gamma_{MAC}} v_F + \frac{F_{cort}}{\gamma_{MAC}} + \frac{F_{cent}}{\gamma_{MAC}}. \tag{2}$$

By fitting such a linear relationship between the velocities of the male-asters complex and the female pronucleus, one can calculate the ratio of their drag coefficient, thus effectively using the female pronucleus as an in vivo measurement probe. To calibrate this probe, we estimated the drag coefficient of the spherical female pronucleus of radius ~3.8 μm using Stokes law and a measured value of cytoplasm viscosity[25], obtaining ~40 pN s μm⁻¹ (Methods). Correcting for average cell boundary effects[13,14], it follows that the drag coefficient of the female pronucleus is ~130 pN s μm⁻¹.

**Centrosome centration exhibits sigmoidal dynamics**. In order to fit the relationship between the velocities of the male-asters complex and female pronucleus, one has to consider whether centering and cortical forces change over time. If these forces are constant over time, their net effect is simply a constant offset in the velocity of the male-asters complex. By contrast, if these forces change over time, the velocity of the male-aster complex

will not only depend on the velocity of the female pronucleus, but also on time.

To explore this question, we first set out to determine the dynamics of centering forces. Centration of the male-asters complex exhibits sigmoidal dynamics in control embryos, with an initial acceleration before pronuclear meeting and a deceleration thereafter[26] (see Fig. 1c, green). Is the initial acceleration due to an interaction with the female pronucleus or do centering forces exhibit sigmoidal dynamics on their own? To address this question, we probed *zyg-12(ct350)* embryos, in which nuclear dynein is depleted and therefore no interaction occurs between asters and pronuclei[20]. To focus strictly on centering forces, we analyzed *zyg-12(ct350)* embryos also depleted of cortical dynein using *goa-1/gpa-16(RNAi)*; in such doubly affected embryos, centrosomes do not separate and move jointly to the cell center (Fig. 2a, b and Supplementary Movie 2, Supplementary Data 1)[24]. This analysis revealed that centrosome velocities exhibit sigmoidal dynamics on a time scale of hundreds of seconds, first increasing and then decreasing when centrosomes approach the cell center (Fig. 2c, d). Therefore, centering forces are not constant and their time variation must be taken into account when fitting Eq. 2 to calculate the drag coefficient of the male-aster complex.

**The force balance driving male-aster complex positioning**. While we could assess the variability of centering forces by depleting cortical and nuclear dynein, we could not likewise assess cortical forces as there is no mutant/RNAi condition to our knowledge that clearly abrogates centering forces. Instead, we set out to analyze pronuclear migration in *goa-1/gpa-16(RNAi)*

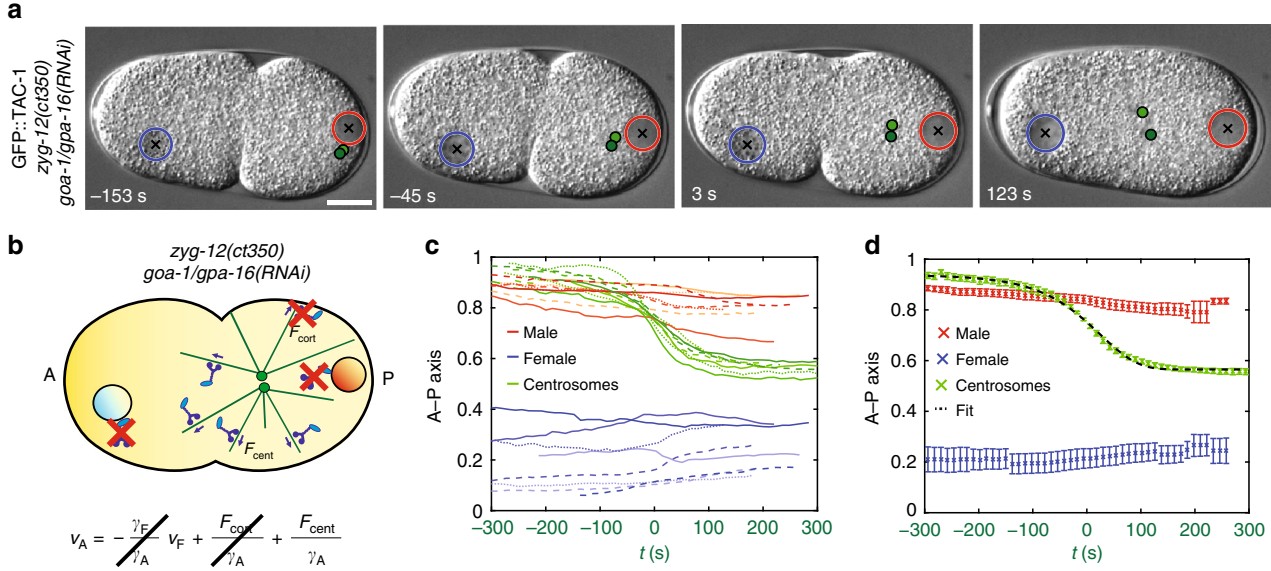

**Fig. 2** Centrosome movements upon depletion of cortical and nuclear dynein reveal centering force dynamics. **a, b** Snapshots and schematics of centrosome centration in *zyg-12(ct350) goa-1/gpa-16(RNAi)* embryos. Since pronuclear meeting does not occur in *zyg-12(ct350) goa-1/gpa-16(RNAi)* embryos, in this figure time 0 s is defined as the half-centration time (indicated by the green lettering on the *x* axis in **c** and **d**; Methods). In **b** of this and subsequent figures, the red crosses represent depleted dynein motors. **c, d** Pronuclei and centrosome midpoint positions along the A–P axis as a function of time in eight *zyg-12(ct350) goa-1/gpa-16(RNAi)* embryos (**c**), as well as their average, represented with S.E.M. **d** Black-dashed line in **d**: fit with sigmoidal model (Eq. 4—Methods, $\chi^2 = 27$, $P = 0.99$)

embryos, which are depleted of cortical dynein, thus removing the potentially confounding effect of cortical forces. We found that the male-asters complex moves at higher velocity toward the anterior during the fast phase of migration in such embryos than in the control, as expected from cortical forces normally partially counteracting anteriorly-directed movements of the male-asters complex (Fig. 3a–d, Supplementary Fig. 2a, Supplementary Movie 3, Supplementary Data 1)[22]. By contrast, the female pronucleus moves at comparable velocities in *goa-1/gpa-16(RNAi)* embryos and in control embryos, as anticipated because cortical forces do not act upon it (Fig. 3d).

We calculated the partial linear correlation between the velocities of the female pronucleus and the male-asters complex in *goa-1/gpa-16(RNAi)* embryos, controlling for time variation, and found that they are indeed correlated, as predicted by Eq. 2 (Fig. 3e). Furthermore, as expected, this correlation is significant 50 s before pronuclear meeting, but not earlier, when the pull exerted on the female pronucleus is negligible (Fig. 3e; slow migration phase $-200 < t < -100$ s, Pearson's partial correlation coefficient $\rho = -0.04$, $P = 0.36$, Student's $t$ test, two-sided). We conclude that the force exerted between the female pronucleus and the male-asters complex contributes significantly to the fast phase of migration.

To further investigate this point, we compared two classes of models, one in which the velocities of the female pronucleus and the male-asters complex are independent (Models 1–3, Supplementary Table 1) and one in which they depend on each other (models 4–9, Supplementary Table 1) (see also Methods). To select the best among these models, we fitted the relationship between the velocities of the male-asters complex and the female pronucleus in each case, evaluating the quality of the fit using the Akaike information criterion (Supplementary Table 1). As indicated above, fitting this relationship requires considering the time variation of centering forces. Since centration occurs on a time scale of hundreds of seconds (Fig. 2d), while the fast phase of pronuclear migration lasts tens of seconds (Fig. 3c), we performed an approximation of the centering forces and

considered models in which centration forces are constant (models 1, 4, and 7), linear (models 2, 5, and 8), or quadratic (models 3 and 6) functions of time, respectively. Furthermore, since the microtubule aster may grow during the fast phase of pronuclear migration, we considered a model in which the drag coefficient of the male-asters complex varies over time (model 9). As reported in Supplementary Table 1, the model selection analysis established that the best model is the one in which the velocity of the male-aster complex depends linearly on the velocity of the female pronucleus (model 5). In this model, centration forces are approximated with a linear function and the drag coefficient is constant over time. Therefore, the velocity of the male-asters complex reads

$$v_{MAC} = -\frac{\gamma_F}{\gamma_{MAC}} v_F + v_0 + mt, \qquad (3)$$

where $m$ is the approximated linear rate of variation of centering forces.

Using this equation, we fitted the relationship between the velocities of the female pronucleus and the male-asters complex in *goa-1/gpa-16(RNAi)* embryos, finding that the drag coefficient of the male-asters complex is $4.4 \pm 0.8$ times that of the female pronucleus (Fig. 3f), in agreement with the theoretical prediction of 3.8[14]. Analogous results were obtained using *gpr-1/2(RNAi)* embryos, in which cortical dynein is also depleted (Supplementary Fig. 3, Supplementary Data 1). Given this ratio and the estimated drag coefficient of the female pronucleus $\gamma_F \sim 130$ pN s μm$^{-1}$, the drag coefficient of the male-asters complex is $\gamma_{MAC} \sim 570$ pN s μm$^{-1}$.

Having estimated this drag coefficient experimentally allowed us to assess the contribution of the different types of forces acting on the male-asters complex during the fast phase of migration (Fig. 3g, Supplementary Fig. 4). From the velocity of female pronuclear migration in embryos depleted of cortical dynein, we deduce that the force exerted between the female pronucleus and the male-asters complex reaches $\sim 0.4\ \gamma_F$ (Fig. 3g—blue; $\sim 50$ pN assuming $\gamma_F \sim 130$ pN s μm$^{-1}$). This value plus the centering force

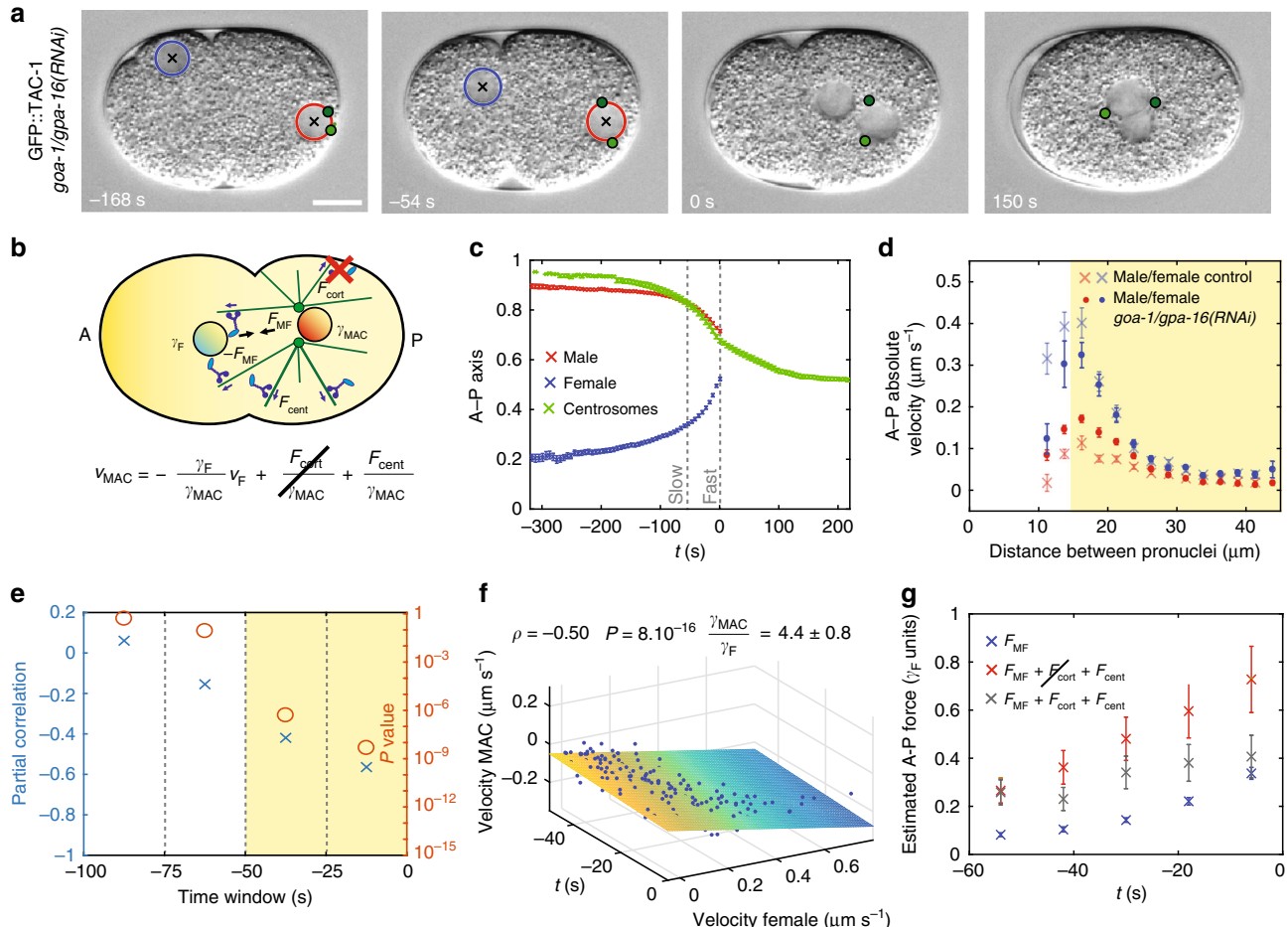

**Fig. 3** Pronuclear migration and centration upon depletion of cortical forces. **a**, **b** Snapshots and schematics of pronuclei and centrosomes in *goa-1/gpa-16(RNAi)* embryos. **c** Average pronuclear and centrosome midpoint positions along the A–P axis as a function of time, with S.E.M. (pronuclei: $n = 31$, centrosomes: $n = 13$). **d** Absolute average pronuclear velocities, with S.E.M., as a function of the distance separating them in the indicated conditions (control, $n = 33$, same as Fig. 1e; *goa-1/gpa-16(RNAi)*, $n = 31$; here and in **e**, **f**, velocities are calculated between successive frames 6 s apart). The velocities during the acceleration phase of the female pronucleus in control and *goa-1/gpa-16(RNAi)* embryos are compatible ($d > 15$ μm, highlighted in yellow; $\chi^2 = 12$; $P = 0.37$), whereas those of the male-asters complex are not ($\chi^2 = 70$; $P < 2 \times 10^{-10}$). **e** Partial Pearson's correlation, controlling for time variation, between pronuclear velocities along the A–P axis over successive time windows ($n = 31$). Here and in other figures, each time point corresponds to a time window of 25 s and the correlated time window ($P < 0.05$) is highlighted in yellow (partial correlation: blue crosses; $P$ value (Student's $t$ test, two-sided): orange circles). **f** Velocity of male pronucleus as a function of time and velocity of the female pronucleus during the correlated phase of pronuclear migration in *goa-1/gpa-16(RNAi)* embryos ($n = 31$, $-50 < t < 0$ s time window). Plane: linear fit $v_{\text{MAC}} = -\frac{\gamma_F}{\gamma_{\text{MAC}}} v_F + v_0 + mt$ ($v_0 = (-0.10 \pm 0.01)$ μm s$^{-1}$; $m = (-0.0008 \pm 0.0003)$ μm s$^{-2}$; errors are S.D.). Here and in Fig. 4f, the Pearson's partial correlation coefficient $\rho$ between the velocities of the male-aster complex (or asters pair) and female pronucleus, controlling for time variation, its $P$ value (Student's $t$ test, two-sided) and the fitted ratio between the drag coefficients of the male-asters complex (or asters pair) and female pronucleus are indicated. **g** Estimated A–P forces (with S.E.M.) acting on the male-asters complex and female pronucleus shortly before their meeting. Blue: force between pronuclei ($n = 31$). Red: sum of forces acting between pronuclei and centering force ($n = 31$). Gray: total force acting on male-asters complex in control embryos ($n = 33$). Here and in Fig. 4g, force is expressed in units of the drag of the female pronucleus, estimated to be ~130 pN s μm$^{-1}$ (Methods)

together correspond to the maximal total force acting on the male-asters complex in embryos depleted of cortical dynein, namely ~0.8 $\gamma_F$ (Fig. 3g—red; ~100 pN assuming $\gamma_F$~130 pN s μm$^{-1}$). In control embryos, these forces are opposed by cortical dynein, so that the peak force is reduced to ~0.4 $\gamma_F$ (Fig. 3g—dark gray; ~50 pN assuming $\gamma_F$~130 pN s μm$^{-1}$). Overall, our analysis allowed us to uncover how different types of forces act on the male-asters complex to direct the fast phase of its migration.

**Estimating centering forces on their own**. To reach a full understanding of the force balance, we set out to estimate the drag coefficient of the asters pair when not attached to the male pronucleus, thus enabling us to directly determine the extent of

centering forces. To this end, we analyzed embryos derived from *top-2(it7)* animals, which lack the male pronucleus, whilst harboring centrosomes[27]. In addition, we depleted cortical dynein using *goa-1/gpa-16(RNAi)* (Fig. 4a, b, Supplementary Movie 4, Supplementary Data 1). Since centrosome separation is driven jointly by cortical and nuclear dynein[24], centrosomes do not separate in *top-2(it7) goa-1/gpa-16(RNAi)* embryos, yet move together toward the cell center (Fig. 4a, c, Supplementary Fig. 5a). After reaching the female pronucleus, centrosomes separate along it (Fig. 4a, 150 s). We found that centrosomes move faster in *top-2 (it7) goa-1/gpa-16(RNAi)* embryos than when the male pronucleus is present, whereas the velocity of the female pronucleus is not altered (Fig. 4d). Furthermore, as predicted by the model in which the female pronucleus contributes to pull the pair of

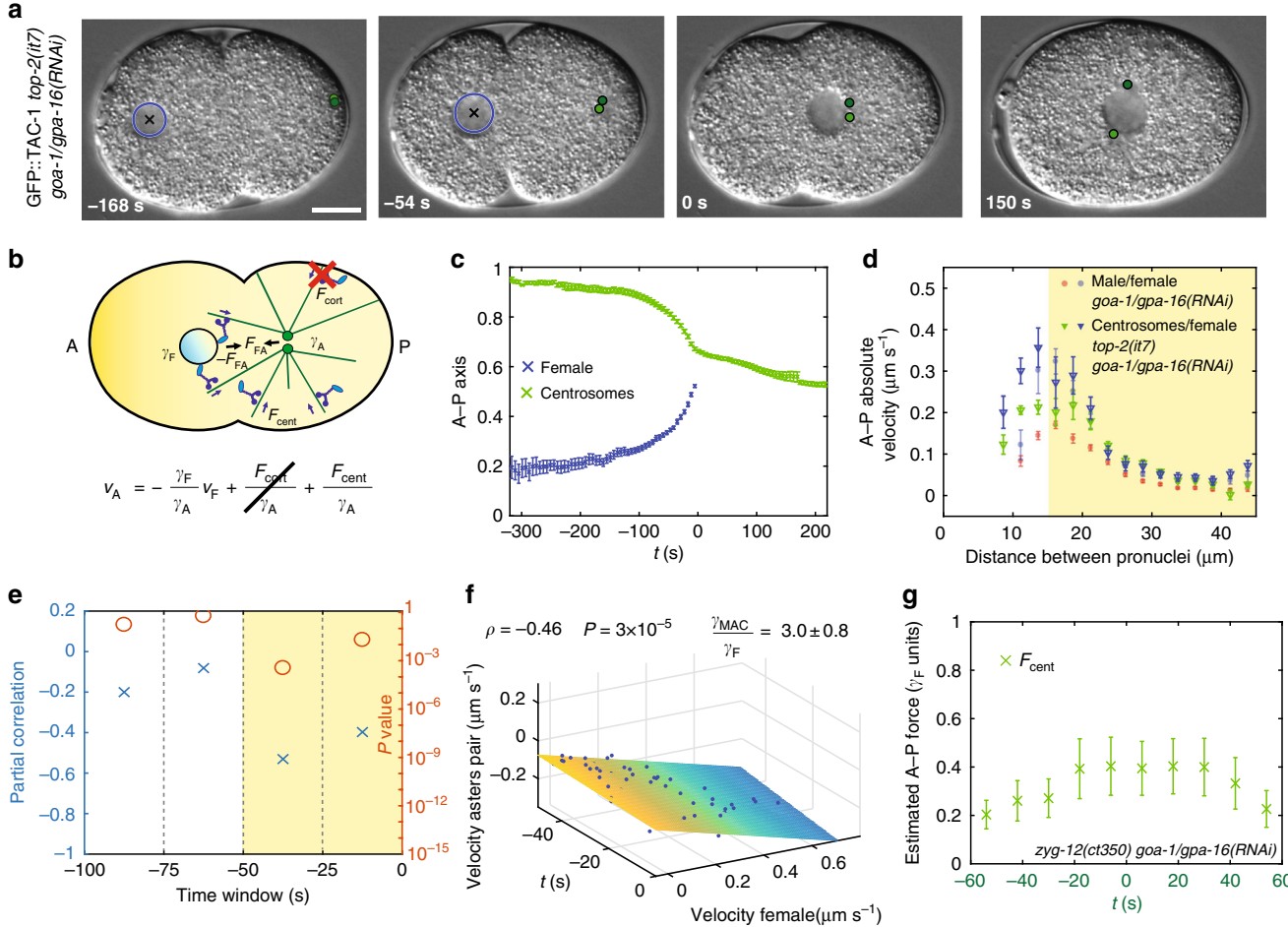

**Fig. 4** Centrosome movements upon depletion of cortical dynein and without male pronucleus uncover drag coefficient of microtubule asters. **a**, **b** Snapshots and schematics of movements of female pronucleus and centrosomes in *top-2(it7) goa-1/gpa-16(RNAi)* embryo. In this figure, time 0 s is at meeting of the asters with the female pronucleus. **c** Average female pronucleus and centrosome midpoint positions along the A–P axis as a function of time, with S.E.M. ($n = 10$). **d** Absolute average velocities, with S.E.M., of the female pronucleus and the centrosome pair as a function of their distance in the indicated conditions (*goa-1/gpa-16(RNAi)*, $n = 31$, same as in Fig. 3d; *top-2(it7) goa-1/gpa-16(RNAi)*, $n = 10$; here and in **e**, **f**, velocities are calculated between successive frames 6 s apart. The velocities of the female pronucleus during the acceleration phase in *goa-1/gpa-16(RNAi)* and *top-2(it7) goa-1/gpa-16(RNAi)* are compatible ($d > 15$ μm, highlighted in yellow; $\chi^2 = 9$; $P = 0.76$), whereas the velocities of the male-asters complex and of the asters pair are not ($\chi^2 = 105$; $P < 1 \times 10^{-16}$). **e** Partial Pearson's correlation, controlling for time variation, between the velocities of the female pronucleus and the two asters along the A–P axis over successive time windows ($n = 10$, partial correlation: blue crosses; $P$ value (Student's $t$ test, two-sided): orange circles; the correlated time window is highlighted in yellow). **f** Velocity of male pronucleus as a function of time and of female pronucleus velocity during the correlated phase of pronuclear migration in *top-2(it7) goa-1/gpa-16(RNAi)* embryos ($n = 10$, $-50 < t < 0$ s time window). Plane: linear fit $v_A = -\frac{\gamma_F}{\gamma_{MAC}}v_F + v_0 + mt$ ($v_0 = (-0.11 \pm 0.04)$ μm s$^{-1}$; $m = (-0.0005 \pm 0.0008)$ μm s$^{-2}$; errors are S.D.). **g** Centering force (with S.E.M., $n = 8$) acting on the asters pair in *zyg-12(ct350) goa-1/gpa-16(RNAi)* embryos as a function of time. Since pronuclear meeting does not occur in *zyg-12(ct350) goa-1/gpa-16(RNAi)* embryos, time 0 s is defined as the half-centration time (indicated by the green lettering on the $x$ axis; Methods)

microtubule asters, centrosomes move faster toward the cell center in *top-2(it7) goa-1/gpa-16(RNAi)* embryos than in *zyg-12 (ct350) goa-1/gpa-16(RNAi)* embryos, in which that pull is absent (Supplementary Fig. 5b, c).

We found also that movements of the female pronucleus are significantly correlated with those of the asters pair in the 50 s before pronuclear meeting in *top-2(it7) goa-1/gpa-16(RNAi)* embryos (Fig. 4e). From the corresponding linear fit of their relationship (Fig. 4f), we determined that the asters pair has a drag coefficient $\gamma_A = 3.0 \pm 0.8$ times that of the female pronucleus, or ~390 pN s μm$^{-1}$. From the calculated asters drag coefficient and from asters velocities in *zyg-12(ct35) goa-1/gpa-16(RNAi)* embryos, we derived the net centering force acting on the two asters around the time of half-centration and found it to increase from ~0.2 $\gamma_F$ to ~0.4 $\gamma_F$ (Fig. 4g; from ~25 pN to ~50 pN assuming $\gamma_F \sim 130$ pN s μm$^{-1}$). This result is compatible with the observation

that centering forces adds up to ~50 pN to the force acting on the male-asters complex in *goa-1/gpa-16(RNAi)* embryo (Fig. 3g), thus providing an integrated understanding of how forces govern migration of asters and pronuclei in the *C. elegans* zygote.

## Discussion

The mechanisms ensuring that microtubule asters are properly positioned within cells constitute an important open question in cell and developmental biology. To understand how microtubules and molecular motors control the position of asters and associated nuclei, we analyzed the reciprocal movements of the male-asters complex and the female pronucleus as they pull on each other during pronuclear migration in *C. elegans*. Our analysis reveals that the translational drag coefficient of the male-asters complex is ~570 pN s μm$^{-1}$ and that of a pair of juxtaposed asters

$\sim$390 pN s $\mu$m$^{-1}$. These drag coefficients include all the potential drag sources, regardless of their nature. Although our findings are compatible with the drag being exerted from cytoplasm viscosity alone[14], other components, such as microtubule polymerization against the cortex, may contribute as well[25,28]. Intriguingly, work using magnetic tweezers to move a microtubule aster in metaphase one-cell *C. elegans* embryos determined the drag coefficient to be $\sim$125 pN s $\mu$m$^{-1}$ per aster[25]. Even if those experiments were performed during mitosis, it is worth noting that the resulting value for a single aster is in the same order of magnitude as half of that measured for two asters here using a completely independent approach.

Interestingly, our measurements show that asters in the *C. elegans* zygote have drag coefficients comparable to, or larger than, that of their main cargo, i.e., the nucleus. Therefore, embryos must cope with the tradeoff between increasing microtubule number to generate stronger microtubule-dependent forces and reducing aster size to achieve smaller frictional drag. This conundrum is even more extreme in larger zygotes, such as those of amphibians or echinoderms, in which the aster is at least an order of magnitude larger than the associated pronucleus, so that aster drag is expected to be much larger than that of the pronuclei[29,30]. Accordingly, the impact of female pronuclear migration on aster movements appears negligible in sea urchin eggs[30].

Estimating drag coefficients allowed us to derive the dynamic contributions of several sources to the force balance driving migration of the male-asters complex and the female pronucleus (Supplementary Fig. 4). First, assuming the drag coefficient of the female pronucleus to be $\gamma_F \sim 130$ pN, the fast migration of the female pronucleus is driven by a pull exerted by the male-asters complex that peaks at $\sim$50 pN (Supplementary Fig. 4a). If this force was exerted by dynein motors working at a stall force of $\sim$6 pN[31], this would correspond to approximately eight to nine active motors. This pull seems to be negligible earlier on, during the slow migration phase, since the movements of the male-asters complex and the female pronucleus are not correlated at that time. Second, a reaction force, equal in magnitude, is exerted on the male-asters complex. Third, the male-asters complex experiences a centering force with sigmoidal dynamics, which peaks at $\sim$50 pN (Supplementary Fig. 4b). This force powers the microtubule asters pair toward the cell center in *zyg-12(ct350) goa-1/goa-16(RNAi)* embryos (Supplementary Fig. 4c). Together with the pull exerted by dynein at the cortex, the balance of forces result in $\sim$50 pN being applied toward the anterior direction onto the male-asters complex, similar to that exerted in the other direction onto the female pronucleus (Supplementary Fig. 4d).

What is the nature of the centering force? One possibility is that the reaction force exerted on microtubules by dynein motors transporting vesicles toward centrosomes generates length-dependent forces that center the microtubule asters[32]. An additional force component may be exerted by dynein held at the cortex by LIN-5, independently of GOA-1/GPA-16 and GPR-1/2[33]. Another possibility is that microtubule polymerization against the cortex pushes asters toward the cell center, a mechanism that has been proposed for maintaining the metaphase spindle in the embryo center a few minutes later[25]. However, microtubule polymerization forces alone do not appear sufficient for centering, since this process is abolished in embryos lacking dynein function, in which microtubules can grow[19].

Overall, our findings provide a robust biophysical framework of aster positioning in the zygote, which is fundamental for understanding the mechanisms governing the union of the two parental genomes. Such analyses can be extended to other systems and contribute to elucidate experimentally how microtubule asters are moved by the cellular force-generating machinery.

## Methods

**Worm strains**. Transgenic worms expressing GFP::TAC-1[34] were maintained at 24 °C. GFP::TAC-1 *zyg-12(ct350)* was maintained at 16 °C and shifted to the restrictive temperature of 24 °C before analysis[24]. The strain KK381 carrying the temperature sensitive allele *top-2(it7)* (previously known as *mel-15(it7)*)[35] was generously provided by Aimée Jaramillo-Lambert and Andy Golden[27], crossed to GFP::TAC-1, maintained at 16 °C and shifted to the restrictive temperature of 24 °C before analysis.

**RNAi**. The RNAi feeding strains for *goa-1/gpa-16(RNAi)* and *gpr-1/2(RNAi)* were described[36]. RNAi was performed by feeding animals as follows: *goa-1/gpa-16 (RNAi) zyg-12(ct350)* by letting adults lay eggs on *goa-1/gpa-16(RNAi)* feeding plates and imaging the progeny of F1 animals after 134–163 h at 16 °C, then 1–4 h at 24 °C; *gpr-1/2(RNAi)* by feeding L1–L2 animals for 40–48 h at 24 °C; *goa-1/gpa-16(RNAi) top-2(it7)* by feeding L1–L2 animals for 44–56 h at 24 °C or L2–L3 animals for 24–32 h at 24 °C. *goa-1/gpa-16(RNAi)* embryos were divided into two groups. One group ($n = 13$) was analyzed previously for the process of centrosome separation that occurs earlier in the cell cycle[24]; in this case, RNAi was performed by letting adults lay eggs on *goa-1/gpa-16(RNAi)* feeding plates and imaging the progeny of F1 animals after 134–163 h at 16 °C, then 1–4 h at 24 °C. For the second group ($n = 18$), RNAi was performed by feeding L1–L2 animals for 44–56 h at 24 °C. We analyzed the two groups separately, observed no phenotypic difference between them, and therefore pooled them. The efficiency of GOA-1/GPA-16 and ZYG-12 depletion was assessed phenotypically as reported[24], that of TOP-2 by the absence of male pronucleus. The efficiency of GPR-1/2 depletion was assessed phenotypically like that of GOA-1/GPA-16 by the absence of spindle oscillations and the impairment of spindle elongation/positioning during mitosis[36]. Note that processes mediated by cortical dynein, including centrosome separation, still occur in embryos depleted of ZYG-12, whereas those relying on nuclear dynein, including pronuclear meeting, still occur in embryos depleted of GOA-1/GPA-16[24].

The embryos analyzed in Figs. 1, 2, Supplementary Fig. 1, as well as a subset of embryos analyzed in Fig. 3, Supplementary Fig. 2a–c ($n = 13/31$) have been used to study the earlier process of centrosome separation[24].

**Imaging**. Embryos were imaged as described[24]. Briefly, embryos were dissected in osmotically balanced culture medium[37] and imaging performed at $24 \pm 0.5$ °C using dual time-lapse DIC and fluorescent microscopy on a Zeiss Axioplan 2 with a $63 \times 1.40$ numerical aperture lens and a 6% neutral density filter to attenuate the 120 W Arc Mercury epifluorescent source. The motorized filter wheel, two external shutters, and the $1392 \times 1040$ pixels 12-bit Photometrics CoolSNAP ES2 camera were controlled by μManager (www.micro-manager.org). Hardware binning 2 was used, resulting in a pixel size of 0.2048 μm. For *goa-1/gpa-16(RNAi) zyg-12(ct350)* embryos, a z-stack of 13 planes 1.5 μm apart was taken every 12 s. In the other conditions, a z-stack of 7 planes 1.5 μm apart was taken every 6 s. The exposure time per plane was 30–100 ms for DIC and 30–60 ms for fluorescence using the Zeiss Filter Set 10. Fluorescence was not used for the second group of *goa-1/gpa-16 (RNAi)* embryos ($n = 18/31$), in which merely pronuclear positions were used for tracking (see above).

**Tracking of centrosomes and pronuclei**. Centrosomes and pronuclei were tracked in 3D as described[24]. In brief, centrosomes were tracked using the Imaris Spot Detection feature (Bitplane) from the onset of centrosome separation until the end of centration/rotation. The position of a centrosome pair was set as the midpoint between the two centrosomes. Pronuclei were tracked in 3D using a custom software written in MATLAB (MathWorks) based on the homogenous appearance of pronuclei with respect to the rough texture of the cytoplasm-containing yolk granules.

**Force balance of pronuclear and aster migration**. We monitored translational movements along the A–P axis to measure the drag coefficient of the microtubule aster, approximated with a rigid body. Eq. 1 is the balance of forces acting on the male-asters complex (asters pair) and female pronucleus along the A-P axis. In particular, the drag force balances the total of the other applied forces, as expected in the highly viscous cytoplasm environment. Translational movements along the A–P axis are assumed to be independent from movements in other directions and from potential rotations. In this case, the A–P component of the drag force is proportional to the A–P velocity and to a characteristic drag coefficient[11]. Such a drag coefficient, in addition to the contribution reflecting the interaction with cytoplasm viscosity, could in principle include contributions from other forces, such as microtubule polymerization forces, if they depend linearly on the velocity of the male-asters complex. Since cytoplasmic flows induced by actomyosin cortical contractions have ceased at pronuclear meeting[38], we consider cytoplasm velocity to be null when the male-asters complex (asters pair) and the female pronucleus are not moving.

**Model selection analysis**. We performed a model selection analysis based on the Akaike information criterion to test the appropriateness of different models in describing the relationship between the velocities of the male-asters complex, the

velocities of the female pronucleus and time (Supplementary Table 1). We considered a set of polynomial models in which the velocity of the male-asters complex depends up to the second power of the velocity of the female pronucleus. Since the centering force varies over a time scale of hundreds of seconds, while the fast migration phase lasts about 50 s, we performed a Taylor expansion of the centering force, up to the second order. Furthermore, we considered a model in which the drag coefficient of the male-asters complex varies over time (Model 9), while velocity of the male-asters complex depends linearly on both time and on the velocity of the female pronucleus. For simplicity, the time variation of the drag coefficient of the male-asters complex (asters pair) is considered linear

$$\gamma_{MAC} = \gamma_{MAC}(0) + rt,$$

where $\gamma_{MAC}(0)$ is the drag coefficient at time 0 s and $r$ is its time variation. Assuming that the variation of the drag coefficient is small during the fast phase of pronuclear migration, we obtain by Taylor approximation

$$V_{MAC} = -\frac{\gamma_F}{\gamma_{MAC(0)} + rt}V_F + V_0 + mt \approx -\frac{\gamma_F}{\gamma_{MAC(0)}}V_F + r't\,V_F + V_0 + mt,$$

where $r' = r\frac{\gamma_F}{\gamma_{MAC(0)}^2}$.

We fitted each polynomial model (fit function—MATLAB) and calculated the likelihood of each model assuming Gaussian errors, therefore calculating the chi-squared of each fit. We evaluated the quality of each model using the Akaike information criterion (AIC—Supplementary Table 1). The selected model, i.e., the model with the lowest AIC, is model 5, in which the velocity of the male-asters complex depends linearly both on the velocity of the female pronucleus and on time, while the drag coefficient of the male-aster complex is constant.

**Calculation of centrosome pair/pronuclei migration curves and velocities**. All data analysis was performed using custom-written MATLAB code. The positions of centrosomes and pronuclei were projected onto the A–P axis and expressed as fractions of A–P axis length. To this end, the eggshell contour was segmented in the middle plane by automatically fitting an ellipse on >20 manually selected points. The A–P axis was defined as the major axis of the fitted ellipses and the posterior side manually selected.

In control and goa-1/gpa-16(RNAi) embryos, the curves of centrosome and pronuclear migration were synchronized with pronuclear meeting as time 0 s. Pronuclear meeting was manually identified generally as the first frame in which at least one of the two centrosomes reaches the female pronucleus.

In zyg-12(ct350) goa-1/gpa-16(RNAi) embryos, each centration curve was fitted with the sigmoidal model

$$x(t) = x_{eq} + (x_{0-}x_{eq})e^{-ce^{dt}}, \tag{4}$$

where $x_0$ and $x_{eq}$ are the initial and final centrosome position, respectively; $c$ and $d$ are two effective parameters. We fitted the sigmoidal model using non linear least squares (fit function—MATLAB). Pronuclear migration and centrosome centration curves were then synchronized between embryos by taking as time 0 s the fitted half-centration time. Synchronized centration curves were averaged and the average curve fitted with the sigmoidal model (Eq. 4).

The velocities of centrosomes and pronuclei were calculated between successive frames. Velocities along the A–P axis were taken positive when directed toward the embryo posterior. The velocity of the male pronucleus was considered as the velocity of the male-asters complex. The velocity of a microtubule asters pair was determined as the velocity of the midpoint between the two centrosomes.

**Linear fitting**. The partial Pearson's correlation coefficient between the velocities of the male-asters complex (or asters pair) and the female pronucleus, controlling for time variation, was calculated in 25s-long time windows. The correlation was considered significant when its P value was <0.05 (Student's t test, two-sided). The relationship between the velocities of the male-asters complex (asters pair) $v_{MAC}$ and the female pronucleus $v_F$ was fitted with Eq. 3 (see main text).

In Eq. 3, the stochastic forces acting on the male-asters complex (asters pair) were modeled by the stochastic offset $v_0$. Such stochasticity, together with potential measurement errors of male-asters complex velocity, does not bias the fit of the slope $\frac{\gamma_F}{\gamma_{MAC}}$, but only increases its statistical error. By contrast, stochastic forces and measurements errors that affect female pronucleus velocity $v_F$ result in a biased slope estimate when ordinary least square regression is used (regression dilution). To avoid this, when calculating the ratio between the drag coefficients of the male-aster complex (asters pair) and female pronucleus, we fitted Eq. 3 using the maximum likelihood estimator for models with errors in both dependent and independent variable and corrected for small sample size ([39]Sec. 2.5.1—the parameter $\alpha$ is taken equal to $n+1$ where $n$ is the number of dependent variables). To estimate the stochastic deviations in female pronucleus velocity, we calculated its distribution during the slow migration phase in goa-1/gpa-16(RNAi) embryos, from 200 to 100 s before pronuclear meeting, when interaction with the asters is non-existent or minimal; such velocities were distributed as a Gaussian (Supplementary Fig. 2b). We estimated the error in the velocity of the female pronucleus to be equal to the S.D. of such Gaussian distribution (0.04 µm s⁻¹). To

fit Eq. 3, the error in time measurements was considered negligible. The errors of the fitted parameters have been calculated by jack-knife resampling[40].

**Estimation of the drag coefficient of the female pronucleus**. We estimated the drag coefficient of the female pronucleus using Stokes law, including the correction for the presence of cell boundaries[14]. The female pronucleus has a radius of ~3.8 µm (Supplementary Fig. 2c). We calculated the viscosity of the cytoplasm from[25], in which magnetic tweezers were used to move µm-sized magnetic beads in the cytoplasm, measuring a drag coefficient of $\gamma_{bead} = 5.3 \pm 1.3$ pN s µm⁻¹. This enabled us to calculate a cytoplasm viscosity of $\eta \sim 0.6 \pm 0.1$ pN s µm⁻² using Stokes law, not considering effects induced by cell boundaries in this case since the size of the beads is almost two orders of magnitudes smaller than their distance from the cell cortex. Stokes law predicts a drag coefficient for the female pronucleus of ~40 ± 8 pN s µm⁻¹. The drag acting on a sphere ~10 µm in diameter is 3.3 times that predicted by Stokes law because of cell boundary effects[14]. Thus, we estimate that the drag coefficient of the migrating female pronucleus is $\gamma_F \sim 130$ pN s µm⁻¹.

**Code availability**. Custom software developed to analyze images and data sets have been written in MATLAB. The software, as well as parameters, are available from the corresponding author on request.

**Data availability**. Centrosome and pronuclear positions are reported in Supplementary Data 1. Additional data sets generated and/or analyzed during the current study are available from the corresponding author on request.

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

## Acknowledgements

We thank Alexandra Bezler, Niccolò Banterle, Antonio Celani, Victoria Deneke, Stefano Di Talia, François Nédélec, and Lukas Von Tobel for advice and/or comments on the manuscript. For strains, we thank Aimée Jaramillo-Lambert and Andy Golden, as well as the *Caenorhabditis* Genetics Center, which is funded by the NIH National Center for Research Resources (NCRR). Supported by the Swiss National Science Foundation (3100A0–122500/1 and 31003A_155942). The funders had no role in study design, data collection and analysis, decision to publish, or preparation of the manuscript.

## Author contributions

A.D. and P.G. designed the project; A.D., A.S., and C.B. conducted the experiments; A.D., A.S., and P.G. analyzed the data; A.D. and P.G. wrote the manuscript.

## Additional information

**Competing interests:** The authors declare no competing financial interests.

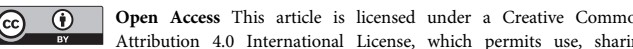

