## [Peer Review File · Nature Communications]

Reviewer #1 (Remarks to the Author):

In this manuscript, De Simone et al. evaluated the ratio of the drag coefficient between the (female pro)nucleus (F) and the complex of (male pro)nucleus and microtubule asters (MAC). This evaluation characterizes the drag of the asters, which have complicated shapes, in vivo for the first time, to my knowledge. The analyses were done by quantifying the velocities of F and MAC in the *C. elegans* embryo. The authors assumed that the velocity of MAC is determined by the contributions of centering forces, cortical forces, and the pulling between MAC and F. Further assuming that the centering forces and cortical forces are constant during the fast phase of pronuclear migration, the authors calculated the ratio of the drag coefficients to be ~ 4 . The authors knocked down several genes to eliminate the contributions of cortical forces, male pronucleus, and/or female pronucleus to dissect the contributions from each component. Given the theoretical estimation of the drag coefficient of F to be 130 [pN s/ μ m], the authors estimated the drag coefficient of MAC to be 470 [pN s/ μ m].

As this is the first characterization of the drag of the microtubule asters in vivo, the manuscript is potentially important to scientists in the field of cell biology. However, the manuscript does not provide strong evidence for its conclusions as follows. Most importantly, it is not clear whether the quantitative data statistically support a significant contribution of F on the movement of MAC (please see comment 1 and 2e,f). Unless the authors clearly overcome these points, I think the manuscript is not acceptable for Nature Communication or other journals.

1. It is not clear from the authors' analyses whether the contribution of the pulling between F and MAC is significant (for MAC migration). I suggest the following analysis to clarify this point. Fit the data (possible *goa-1/gpa-16* RNAi data) to other equations than eqs. 2 and 3. For the equations, I suggest $v_{MAC} = v_0$, $v_{MAC} = v_0 + mt$, and $v_{MAC} = v_0 + mt + nt^2$. Then evaluate the relative quality of these 5 equations/models with information criterion such as AIC, BIC etc. If the model with vF term is selected, the critical assumption of this study will be supported.

2. The authors used data from $[-50s < t < -25s]$ for Fig. 1(g) and $[-50s < t < -0s]$ for Figs. 2(f) and 4(f).

2-a) In Figs. 2f and 4f, I assume the authors calculate the average velocities of $[-50s < t < -25s]$ and $[-25s < t < 0s]$ for each dataset independently and plotted. Am I correct? Otherwise, did the authors calculate the average velocity of $[-50s < t < -0s]$? If the latter is the case, I do not think it is

appropriate because the velocity of $[-50s < t < -25s]$ and $[-25s < t < 0s]$ are dramatically different in many cases.

2-b) The authors described that “we will consider the cortical and centering forces to be independent of time. We will later test the validity of this approximation by depleting cortical forces and considering the time-variation of centering forces. (P.7, L.18-)”. I think the authors are referring to Figs. 2g and 4g, but I could not find clear discussion on this point. Instead, I found difference of the centering forces between $[-50s < t < -25s]$ and $[-25s < t < 0s]$ in these panels. Why can the authors argue the centering forces are constant?

2-c) If the centering forces are not constant between $[-50s < t < -25s]$ and $[-25s < t < 0s]$, which I think the authors showed in Figs. 2g and 4g, it is not appropriate to mix the data from $[-50s < t < -25s]$ and $[-25s < t < 0s]$ in Figs. 2f and 4f to calculate the ratio between the gammas of MAC/A and F. The authors should only use $[-50s < t < -25s]$ data to be consistent with Fig. 1g.

2-d) What is the rationale for assuming cortical forces to be constant? From Fig. 2g, I find cortical forces to increase from $[-50s < t < -25s]$ to $[-25s < t < 0s]$.

2-e) To consider whether the movement of the asters are affected significantly by the female pronucleus during the $[-50s < t < -25s]$ period (related to comment 1), the comparison between the measurements in Figs. 3 and 4 should be useful. I am afraid that the trajectories of the centrosomes in Fig. 3d and Fig. 4c during $[-50s < t < -25s]$ period (of Fig. 4c) overlap with each other, which indicates the contribution of the female pronucleus is not significant at this time period.

2-f) From a similar viewpoint, I request the authors to make histograms of the velocities of the centrosomes in the experiments of Fig. 3 and Fig. 4. I am again afraid that the velocity from Fig. 4 during the $[-50s < t < -25s]$ period is indistinguishable from that from Fig. 3, which again indicates the contribution of the female pronucleus is not significant at this time period.

3. In Figs. 2g and 4g, the vertical axis shows the force in $[\mu\text{N}]$. The value is based on the assumption that $\gamma\text{-F}$ is $130 [\mu\text{N s}/\mu\text{m}]$. As the other data in this manuscript is based solely on the experimental measurements, I feel it is misleading to include the theoretical assumption to present the experimental data here. In fact, in the other place, the authors describe another value as the estimate for $\gamma\text{-F}$ ($40 [\mu\text{N s}/\mu\text{m}]$, P. 8, L.5), which is $1/3$ of the other value. I recommend using

different unit for Figs. 2g and 4g. For example, to show the force in an arbitrary unit using $\gamma \cdot F = 1$. I felt using [pN] units in the discussion is acceptable and informative.

4. P.10, L. 14-16: "Overall, our analysis reveals that ...". What the authors concluded here was the assumption of the analysis (P.7, eq. (1)). I think this is logically inappropriate.

5. P.13, L2: "v25"? Is it typo?

Reviewer #2 (Remarks to the Author):

This paper advances our understanding of how microtubule asters move in zygotes. This basic biology topic is of significant interest to embryologists and physical biologists, and suited to the journal. The paper is generally interesting and well executed and could be accepted in its current form. We have only minor questions that could be addressed without the need for new experiments.

By imaging the motions of centrosomes and female/male pronuclei in *C. elegans* zygotes depleted of dynein on the cortex and/or nuclear envelope, the authors quantified forces on and drag coefficients of asters and pronuclei, without directly measuring the forces, e.g., using magnetic tweezers. Furthermore, by imaging the motions of centrosomes in zygotes without male pronuclei, they isolated how astral microtubules alone contribute to the drag coefficient. This measurement is important because recent simulations predict that hydrodynamic interactions between microtubules are significant, so the microtubule aster behaves as a porous shell (Nazockdast et al. 2017 - cited by the authors). I recommend the paper to be accepted without additional experiments.

Though not big concerns, I was confused by two related points:

Point 1: What powers the fast migration phase?

In Figure 3D, they show that inhibition of dynein on both the cortex and nuclear envelope still results in sigmoidal dynamics with a fast migration phase. This seems inconsistent with an earlier claim, on page 8, where they note, "These results provide a quantitative confirmation of the model in which the male-asters complex pulls on the female pronucleus to power the fast migration phase." The centrosome trajectories in Figures 2C and 4C (with dynein on the nuclear envelope) do exhibit a kink where they meet the female pronuclei, consistent with the claim on page 8, but otherwise the trajectories appear similar to those in Figure 3D (without dynein on the nuclear envelope) - so it seems cytoplasmic anchors other than the female pronucleus may contribute more to the fast migration phase. This is not a big concern because it is not one of their main claims.

Related to this question, they comment on the nature of centering forces in the second to last paragraph of the discussion on page 15. The aforementioned simulations predict that cytoplasmic flows can be used to infer the nature of centering forces (Nazockdast et al. 2017). Looking at their DIC movies, I wonder if they can characterize cytoplasmic flows based on the motions of smaller particles distributed throughout the cytoplasm. In Figure 3D, I also note slight inward drift of male and female pronuclei, perhaps related to these cytoplasmic flows. In a previous paper, the first author has quantified cortical actomyosin flows related to centrosome separation (De Simone et al. 2016).

Point 2: More generally, how do the authors deal with time variation?

In all experiments, the centrosomes exhibit sigmoidal dynamics, with a slow migration phase preceding a fast migration phase. To model their experiments, as an approximate first step, they assume all forces and drag coefficients are constant, which makes sense to me as a first step.

To account for time variation in a second step, they allow centration forces (but not the drag coefficient of the aster - see subpoint 2.1) to vary with time. I am confused by their Taylor expansion in Equation 3 on page 11. I assume they expanded around $t = 0$, the half-centration time based on fitting an asymmetric sigmoidal model to the trajectories (see subpoint 2.2). But for symmetric sigmoids, velocity is maximal at the half-centration time, in which case the first-order term in the Taylor expansion should be 0 (since the first derivative at a local maximum is 0)? If I have misunderstood, perhaps the authors could address this confusion by motivating why the forces should vary linearly with time around $t = 0$, and if so, if the forces should linearly increase or decrease with time? This is not a big concern because their analysis that accounts time variations gives a similar result as their analysis that ignores time variations.

Subpoint 2.1: Why do they assume force varies with time, but the drag coefficient does not vary with time?

Though force and drag are both complicated functions of aster size and structure, I suppose the drag depends more on aster size, whereas the force depends more on aster structure, in particular the (a)symmetry of the aster. Thus, I anticipate that as the aster grows both the drag and force increase, then as the aster centers the drag remains high but the force decreases - though of course both are complicated. So my interpretation is the authors assume the drag coefficient does not vary with time because they are considering the fast migration phase, in which the aster size does not change?

Subpoint 2.2: Why does their sigmoidal model have that functional form?

This is not a big concern because their estimates of drag coefficients were based on linear regression, and I think they used the sigmoidal model just to synchronize trajectories. I assume they chose this functional form with two parameters (in their notation, c and d) to deal with asymmetries, rather than a symmetric sigmoid with just one parameter.

Not necessarily suggesting for this paper, but I am curious if they could derive a sigmoidal model based on a toy model, such as in a recent paper on aster centering in sea urchin eggs (Tanimoto et al. 2016 - cited by the authors), which predicts slow and fast migration phases and large aster asymmetry in a low force regime, consistent with the 6 or 7 active dynein motors estimated in this paper. If so, perhaps the authors could incorporate time variation in a more natural way?

Reviewer #3 (Remarks to the Author):

The manuscript by De Simone and coworkers reports an elegant

experimental and theoretical analysis of the forces that position nuclei and asters in *C. elegans* zygotes. The positioning of intracellular structures and organelles is important for proper functioning and division of cells. While various forces inside cells are involved, organelle motion strongly depends on the drag generated by cell cytoplasm. Despite some recent advances in force measurement inside cells, little has been known about the cytoplasmic drag on different intracellular structures due to their geometric complexity

(witness the centrosomal MT array) and the confinement of the

cell. That said, recent advances in modeling and computation have allowed estimates of the influence of cellular confinement and associated MT arrays upon drag coefficients of nuclei.

De Simone et al use quantitative microscopy, genetic

perturbations, and a simple organizing force-balance model to assess and measure the drag and force couplings on the centrosomal MT arrays and the cell nuclei during first cell division in *C. elegans* embryos. In particular they establish that the female pronucleus and male pronuclear complex (which includes the centrosomal arrays) have velocities in register as they approach each other. They use this observation to estimate the ratio of their drag coefficients, and find this is in close agreement with the new theoretical estimates of Nazockdast et al (2017) that take into account cellular confinement and MT array drag. The authors then used this data, together with genetic perturbations that selectively remove various sources of force and coupling, to estimate forces on the pronuclei as they migrate towards the cell center. These forces they ascribe to cortically bound dynein force generators, and to an additional but sub-dominant "centering force" (e.g. cytoplasmic dyneins pulling upon centrosomal MTs).

The paper is well-organized, concise and well-written, the data is of high quality, and the results are important and interesting. I am happy to recommend publication in Nature Communications. However, a few issues should be addressed before publication and can improve the study:

1: The authors mentioned that cortical forces are generated by the GOA-1/GPA-16, GPR-1/2, and LIN-5 complexes, and conjecture that the centering forces are caused by cytoplasmic dynein. It is worth mentioning other sources of pulling forces as described in Gusnowski and Srayko, JCB 2011, which can act as centering forces but are independent of GOA and GPR pathways.

2: There are various mutations that can significantly change the size of the pronuclei in *C. elegans* embryos. It would be

interesting to measure the speed of pronuclei migration in these mutants and compare it to the prediction of the model knowing the size of the pronuclei and the drag.

3: Following the theoretical study in (14) by Nazockdast and

coworkers, there are two major drag coefficients associated with the motion of pronuclei: translational drag and rotational drag. While the authors well characterized the translational drag, they can perform a similar analysis to estimate the rotational drag as well. This can be an interesting addition to this paper, or as a future continuation of this study. (I see that the paper by Nazockdast et al is now published in MBOC. Hence, the authors should update that citation).

Point-by-point response to the reviewers' comments

We thank the reviewers for their appreciation of our work, as well as for having raised a number of important points, which we addressed in full as detailed below point-by-point.

Reviewer #1 (Remarks to the Author):

In this manuscript, De Simone et al. evaluated the ratio of the drag coefficient between the (female pro)nucleus (F) and the complex of (male pro)nucleus and microtubule asters (MAC). This evaluation characterizes the drag of the asters, which have complicated shapes, in vivo for the first time, to my knowledge. The analyses were done by quantifying the velocities of F and MAC in the *C. elegans* embryo. The authors assumed that the velocity of MAC is determined by the contributions of centering forces, cortical forces, and the pulling between MAC and F. Further assuming that the centering forces and cortical forces are constant during the fast phase of pronuclear migration, the authors calculated the ratio of the drag coefficients to be ~ 4 . The authors knocked down several genes to eliminate the contributions of cortical forces, male pronucleus, and/or female pronucleus to dissect the contributions from each component. Given the theoretical estimation of the drag coefficient of F to be 130 [pN s/um], the authors estimated the drag coefficient of MAC to be 470 [pN s/um].

As this is the first characterization of the drag of the microtubule asters in vivo, the manuscript is potentially important to scientists in the field of cell biology. However, the manuscript does not provide strong evidence for its conclusions as follows. Most importantly, it is not clear whether the quantitative data statistically support a significant contribution of F on the movement of MAC (please see comment 1 and 2e,f). Unless the authors clearly overcome these points, I think the manuscript is not acceptable for Nature Communication or other journals.

> We thank the Reviewer for her/his positive assessment of the importance of our work, as well as for the constructive comments. As described in detail below, we have performed the suggested quantitative analysis, which supports the model in which the pull exerted between the male-asters complex and the female pronucleus contributes significantly to pronuclear migration. Considering this important result, we have decided to not utilize the more approximated models (Eq. 2 in the previous manuscript) and rather to directly report the results obtained fitting with Eq. 3. As a result, we have reorganized the text and rearranged the order of Figures.

1. It is not clear from the authors' analyses whether the contribution of the pulling between F and MAC is significant (for MAC migration). I suggest the following analysis to clarify this point. Fit the data

(possible goa-1/gpa-16 RNAi data) to other equations than eqs. 2 and 3. For the equations, I suggest $v_{MAC} = v_0$, $v_{MAC} = v_0 + mt$, and $v_{MAC} = v_0 + mt + nt^2$. Then evaluate the relative quality of these 5 equations/models with information criterion such as AIC, BIC etc. If the model with vF term is selected, the critical assumption of this study will be supported.

> We thank the Reviewer for making this critical suggestion. We conducted this analysis, considering nine models all together, including the ones mentioned by the Reviewer, and applied the AIC to compare them. Importantly, this analysis revealed that the model in which the velocity of the male-asters complex depends linearly on the velocity of the female and on time best describes our dataset. The results of the model selection analysis are described extensively in the main text (p. 10), in the Methods (p. 30-31) and reported also in the new Supplementary Table 1.

2. The authors used data from $[-50s < t < -25s]$ for Fig. 1(g) and $[-50s < t < -0s]$ for Figs. 2(f) and 4(f).

> The choice of using the time-window $[-50 s < t < -25s]$ to fit the control dataset (Fig. 1) and the time-window $[-50 s < t < 0 s]$ to fit all other Figures was motivated by the results of the correlation analysis presented in Fig. 1e, 3e, 4e and Fig. S3b of the original manuscript. As described in the response to point 2-c below, although the results of the original Fig. 1g were in accordance with the rest of manuscript, they were based on the assumption of constant cortical forces, which is difficult to test experimentally. Owing to this uncertainty, we have decided not to include this panel in the revised manuscript, also because it merely served as an introduction to the critical results reported in Figures 2-4. Therefore, data from the $[-50 s < t < 0 s]$ time-window is considered throughout the revised manuscript to estimate the drag coefficient of the male-asters complex (or asters pair in the absence of male pronucleus).

2-a) In Figs. 2f and 4f, I assume the authors calculate the average velocities of $[-50s < t < -25s]$ and $[-25s < t < 0s]$ for each dataset independently and plotted. Am I correct? Otherwise, did the authors calculate the average velocity of $[-50s < t < -0s]$? If the latter is the case, I do not think it is appropriate because the velocity of $[-50s < t < -25s]$ and $[-25s < t < 0s]$ are dramatically different in many cases.

> We apologize for not having explained well enough how the average velocities were calculated. In fact, each time-point corresponds to the velocity of the male-asters complex and female pronucleus calculated between successive frames (6 s apart) in the $[-50 s < t < 0 s]$ time-window, and not to averages over longer time-windows. We now clarify this point in the Legends of Fig. 3, 4 and of Supplementary Fig. 3 (p. 23-25 and p. 3 of the Supplementary Material, respectively), as well as in the Methods (p. 31-32).

2-b) The authors described that “we will consider the cortical and centering forces to be independent of time. We will later test the validity of this approximation by depleting cortical forces and considering the

time-variation of centering forces. (P.7, L.18-)''. I think the authors are referring to Figs. 2g and 4g, but I could not find clear discussion on this point. Instead, I found difference of the centering forces between $[-50s < t < -25s]$ and $[-25s < t < 0s]$ in these panels. Why can the authors argue the centering forces are constant?

2-c) If the centering forces are not constant between $[-50s < t < -25s]$ and $[-25s < t < 0s]$, which I think the authors showed in Figs. 2g and 4g, it is not appropriate to mix the data from $[-50s < t < -25s]$ and $[-25s < t < 0s]$ in Figs. 2f and 4f to calculate the ratio between the gammas of MAC/A and F. The authors should only use $[-50s < t < -25s]$ data to be consistent with Fig. 1g.

> The Reviewer points out correctly that centering forces are variable over time and, therefore, that time variation should be considered when fitting the relationship between the velocities of male-asters complex and female pronucleus. Accordingly, the model selection analysis suggested by this Reviewer confirms that the model including a linear time-variation of centering forces (Model 5 – Eq. 3) describes better our dataset than the ones not including it. As stated above, in the revised manuscript, we consider directly the time-variation of centering forces and fit our dataset using Eq. 3. Since we now consider explicitly the time-dependency of centering forces, we think it is appropriate to pool data from the $[-50 s < t < -25 s]$ and $[-25 s < t < 0]$ time-windows. These considerations are reported in the main text and Methods of the revised manuscript (p. 10, 30-31), as well as in the new Supplementary Table 1. See also response to point 2 of Reviewer 2.

2-d) What is the rationale for assuming cortical forces to be constant? From Fig. 2g, I find cortical forces to increase from $[-50s < t < -25s]$ to $[-25s < t < 0s]$.

> The Reviewer wonders whether it was appropriate to assume constant cortical forces in Fig. 1g of the original manuscript. There, we had performed the fit as a first approximated introductory step, before proceeding with more refined models and the analysis upon cortical force depletion. Even if the drag calculated assuming constant cortical forces in control embryos is compatible with that calculated in embryos depleted of cortical dynein, we now realize that this initial introductory step could be confounding for the reader. Given this consideration, we decided not to include Fig. 1g in the revised manuscript and instead to focus right away on embryos depleted of cortical dynein.

2-e) To consider whether the movement of the asters are affected significantly by the female pronucleus during the $[-50s < t < -25s]$ period (related to comment 1), the comparison between the measurements in Figs. 3 and 4 should be useful. I am afraid that the trajectories of the centrosomes in Fig. 3d and Fig. 4c during $[-50s < t < -25s]$ period (of Fig. 4c) overlap with

each other, which indicates the contribution of the female pronucleus is not significant at this time period.

> We thank the Reviewer for bringing up this important point. Prompted by her/his suggestion, we compared centrosome trajectories in *zyg-12(ct350) goa-1/gpa-16(RNAi)* and *top-2(it7) goa-1/gpa-16(RNAi)* embryos during the fast phase of pronuclear migration, when the pull between the microtubule asters and the female pronucleus is expected to be most relevant. Importantly, this analysis established that centrosomes are significantly faster in *top-2(it7) goa-1/gpa-16(RNAi)* embryos, in which the microtubule aster pair is expected to be pulled by the female toward the cell center, than in *zyg-12(ct350) goa-1/gpa-16(RNAi)*, in which centrosomes are not expected to interact with pronuclei (z-test: $P = 0.007$). This analysis is now reported in the main text (p. 12) and in the new Supplementary Fig. 5b-c.

2-f) From a similar viewpoint, I request the authors to make histograms of the velocities of the centrosomes in the experiments of Fig. 3 and Fig. 4. I am again afraid that the velocity from Fig. 4 during the $[-50s < t < -25s]$ period is indistinguishable from that from Fig. 3, which again indicates the contribution of the female pronucleus is not significant at this time period.

> We thank the Reviewer for this suggestion. We have calculated the velocities in the $[-50s < t < 0s]$ time window now used throughout the manuscript, finding that centrosomes are faster in *top-2(it7) goa-1/gpa-16(RNAi)* embryos, when the microtubule aster pair is expected to be pulled by the female toward the cell center, than in *zyg-12(ct350) goa-1/gpa-16(RNAi)*, in which centrosomes are expected to not interact with pronuclei (Kolmogorov-Smirnov test: $P = 0.01$). Although interesting in principle, we have decided to not include this result in the revised manuscript since it seems partially redundant with the new Supplementary Fig. S5, but could of course do so if the Reviewer or the Editor thought otherwise. We include the requested histogram for the Reviewer's consideration below. See also response to point 1 of Reviewer 2.

3. In Figs. 2g and 4g, the vertical axis shows the force in [pN]. The value is based on the assumption that γ -F is 130 [pN s/um]. As the other data in this manuscript is based solely on the experimental measurements, I feel it is misleading to include the theoretical assumption to present the experimental data here. In fact, in the other place, the authors describe another value as the estimate for γ -F (40 [pN s/um], P. 8, L.5), which is 1/3 of the other value. I recommend using different unit for Figs. 2g and 4g. For example, to show the force in an arbitrary unit using γ -F = 1. I felt using [pN] units in the discussion is acceptable and informative.

> Thank you for making this useful suggestion. As a result, we have changed the Figures and now report force in γ -F units, apart from the discussion section, as suggested.

4. P.10, L. 14-16: "Overall, our analysis reveals that ...". What the authors concluded here was the assumption of the analysis (P.7, eq. (1)). I think this is logically inappropriate.

> We thank the Reviewer for her/his comment. We have modified the text in the revised manuscript to assert instead that our analysis assesses the balance of forces acting during the fast phase of pronuclear migration (p. 11).

5. P.13, L2: "v25"? Is it typo?

Indeed -we thank the Reviewer for having spotted this typo that has been corrected in the revised manuscript (page 12).

Reviewer #2 (Remarks to the Author):

This paper advances our understanding of how microtubule asters move in zygotes. This basic biology topic is of significant interest to embryologists and physical biologists, and suited to the journal. The paper is generally interesting and well executed and could be accepted in its current form. We have only minor questions that could be addressed without the need for new experiments.

By imaging the motions of centrosomes and female/male pronuclei in *C. elegans* zygotes depleted of dynein on the cortex and/or nuclear envelope, the authors quantified forces on and drag coefficients of asters and pronuclei, without directly measuring the forces, e.g., using magnetic tweezers. Furthermore, by imaging the motions of centrosomes in zygotes without male pronuclei, they isolated how astral microtubules alone contribute to the drag coefficient. This measurement is important because recent simulations predict that hydrodynamic interactions between microtubules are significant, so the

microtubule aster behaves as a porous shell (Nazockdast et al. 2017 - cited by the authors). I recommend the paper to be accepted without additional experiments.

> We thank the Reviewer for her/his positive assessment of our work. We have address the outstanding points as delineated below.

Though not big concerns, I was confused by two related points:

Point 1: What powers the fast migration phase?

In Figure 3D, they show that inhibition of dynein on both the cortex and nuclear envelope still results in sigmoidal dynamics with a fast migration phase. This seems inconsistent with an earlier claim, on page 8, where they note, "These results provide a quantitative confirmation of the model in which the male-asters complex pulls on the female pronucleus to power the fast migration phase." The centrosome trajectories in Figures 2C and 4C (with dynein on the nuclear envelope) do exhibit a kink where they meet the female pronuclei, consistent with the claim on page 8, but otherwise the trajectories appear similar to those in Figure 3D (without dynein on the nuclear envelope) - so it seems cytoplasmic anchors other than the female pronucleus may contribute more to the fast migration phase. This is not a big concern because it is not one of their main claims.

> We agree that centering forces are at least as strong as the pull exerted between the male-asters complex and the female pronucleus, as is evident from the revised Fig. 3g and 4g. How is this compatible with centration curves being qualitatively similar when centrosomes are not pulled by the female pronucleus as in *zyg-12(ct350) goa-1/gpa-16(RNAi)* embryos? We think this is because centrosomes interact with the female pronucleus only during the fast migration phase. Therefore, during most of the time that centrosomes move, forces are dominated by the centration force reported in Fig. 4g. Instead, during the fast migration phase, the pull exerted between the microtubule asters and the female pronucleus increases centrosome velocities. We now mention this point explicitly in the Discussion of the revised manuscript (p. 14). Furthermore, we modified the sentence singled out by the Reviewer to clarify that the pull between the male-asters complex and female pronucleus contributes to, but does not power entirely, the fast migration phase (p. 9). See also response to point 2-f of Reviewer 1.

Related to this question, they comment on the nature of centering forces in the second to last paragraph of the discussion on page 15. The aforementioned simulations predict that cytoplasmic flows can be used to infer the nature of centering forces (Nazockdast et al. 2017). Looking at their DIC movies, I wonder if they can characterize cytoplasmic flows based on the motions of smaller particles distributed throughout the cytoplasm. In Figure 3D, I also note slight inward drift of male and female pronuclei, perhaps related

to these cytoplasmic flows. In a previous paper, the first author has quantified cortical actomyosin flows related to centrosome separation (De Simone et al. 2016).

> The Reviewer suggests an interesting analysis to infer the nature of the centering forces and to distinguish between forces exerted along microtubules or at microtubule tips touching the cortex (Nazockdats et al., MBoC, PMID: 28331070). However, this analysis requires the automatized quantification of yolk granule movements and their comparison with the flow profiles predicted by the simulations presented in (Nazockdast et al., MBoC, PMID: 28331070), something that is not trivial to achieve, as it would notably require developing a new experimental and computational methodology. We think that such an analysis, whilst interesting, goes well beyond the scope of the present manuscript.

Point 2: More generally, how do the authors deal with time variation?

In all experiments, the centrosomes exhibit sigmoidal dynamics, with a slow migration phase preceding a fast migration phase. To model their experiments, as an approximate first step, they assume all forces and drag coefficients are constant, which makes sense to me as a first step.

To account for time variation in a second step, they allow centration forces (but not the drag coefficient of the aster - see subpoint 2.1) to vary with time. I am confused by their Taylor expansion in Equation 3 on page 11. I assume they expanded around $t = 0$, the half-centration time based on fitting an asymmetric sigmoidal model to the trajectories (see subpoint 2.2). But for symmetric sigmoids, velocity is maximal at the half-centration time, in which case the first-order term in the Taylor expansion should be 0 (since the first derivative at a local maximum is 0)? If I have misunderstood, perhaps the authors could address this confusion by motivating why the forces should vary linearly with time around $t = 0$, and if so, if the forces should linearly increase or decrease with time? This is not a big concern because their analysis that accounts time variations gives a similar result as their analysis that ignores time variations.

> The Reviewer asks if centering forces have been approximated by Taylor expansion around $t = 0$ s and, more generally, how the dynamics of centering forces has been approximated. It appears that the Reviewer has been misled by the fact that in the previous Fig. 3 (Fig. 2 in the revised manuscript), $t = 0$ s is not the time of pronuclear meeting, as is the case in the rest of the manuscript. Indeed, pronuclei do not meet in *zyg-12(ct350) goa-1/gpa-16(RNAi)* embryos, such that here $t = 0$ s is defined as the half-centration time. In control embryos and in other mutant/RNAi conditions in which pronuclei meet, the time of pronuclear meeting is not necessarily close to the half-centration time and, therefore, the centering force does not have necessarily a null derivative. We apologize for not having been sufficiently clear in the initial submission concerning this point, which is now better explained in the legend of Fig. 2 and 4 (p. 23,

25-26), and by showing the lettering of the x-axis with a different color in Fig. 2c, 2d and 4g to further emphasize this point in a graphical fashion.

The Reviewer also wonders why centering forces have been approximated to the first order, i.e. linearized. In the revised manuscript, we have performed a model selection analysis based on the Akaike Information Criterion to address this point in full (p. 10, 30-31, as well as in the new Supplementary Table 1). We compared models in which the centering force is constant, varying linear or quadratically over time. This analysis revealed that the model with a linearly increasing centering force is the best among those considered to describe our dataset. Therefore, in the revised manuscript, we decided not to use the more approximated models, but instead to fit directly the dataset using Eq. 3, which includes the time variation of centering forces (see pages 10-12 of the revised manuscript and new Supplementary Table 1). As a result, Fig. 2 and 3 of the original manuscript have been swapped. See also response to point 2-b, -c of Reviewer 1.

Subpoint 2.1: Why do they assume force varies with time, but the drag coefficient does not vary with time? Though force and drag are both complicated functions of aster size and structure, I suppose the drag depends more on aster size, whereas the force depends more on aster structure, in particular the (a)symmetry of the aster. Thus, I anticipate that as the aster grows both the drag and force increase, then as the aster centers the drag remains high but the force decreases - though of course both are complicated. So my interpretation is the authors assume the drag coefficient does not vary with time because they are considering the fast migration phase, in which the aster size does not change?

> The Reviewer asks why the drag coefficient of the male-asters complex is assumed to be constant. As the Reviewer notes, when the drag of the aster is expected to increase as the aster becomes larger. In *C. elegans*, microtubules grow at a speed of $\sim 0.7\text{-}1 \mu\text{m s}^{-1}$, so that they can span the entire embryo length in $\sim 50\text{-}70$ s. Since the fast migration phase starts several hundreds of seconds after aster nucleation onset (see Fig. 1d, 4c) (Srayko M et al., Dev Cell, 2005, PMID: 16054029), the distribution of microtubule lengths should be at quasi-steady state by then. However, centrosomes grow in size throughout the whole centration phase, resulting in increased microtubule numbers (Decker M. et al., Curr Bio, 2011, PMID: 21802300; Zwicker D. et al, PNAS, 2014, PMID: 24979791), which may indeed increase the drag coefficient. Importantly, however, the fast phase of migration last only ~ 50 s, so that the drag coefficient should not grow considerably during this time. We have addressed this point quantitatively in the revised manuscript using a model in which the drag coefficient increases linearly over time (Model 9 in Supplementary Table 1). Fitting the relationship between velocities of the male-asters complex, the female pronucleus and time, we found by the Akaike Information Criterion that this model has a lower

quality to describe our data than if the drag coefficient of the male-asters complex is kept constant. This important novel analysis is reported in the revised manuscript (p. 10, 30-31 and Supplementary Table 1) and demonstrates that the approximation of constant drag coefficients is appropriate during the fast migration phase.

Subpoint 2.2: Why does their sigmoidal model have that functional form?

This is not a big concern because their estimates of drag coefficients were based on linear regression, and I think they used the sigmoidal model just to synchronize trajectories. I assume they chose this functional form with two parameters (in their notation, c and d) to deal with asymmetries, rather than a symmetric sigmoid with just one parameter.

Not necessarily suggesting for this paper, but I am curious if they could derive a sigmoidal model based on a toy model, such as in a recent paper on aster centering in sea urchin eggs (Tanimoto et al. 2016 - cited by the authors), which predicts slow and fast migration phases and large aster asymmetry in a low force regime, consistent with the 6 or 7 active dynein motors estimated in this paper. If so, perhaps the authors could incorporate time variation in a more natural way?

> The Reviewer asks whether we could develop a simple mathematical model of centration. Prompted by this suggestion, we have done so and report this in an Appendix at the end of the present document.

However, as also mentioned by the Reviewer, we decided not to include this model in the manuscript *per se* since the sigmoidal Eq. 4 is merely used to synchronize curves.

Reviewer #3 (Remarks to the Author):

The manuscript by De Simone and coworkers reports an elegant experimental and theoretical analysis of the forces that position nuclei and asters in *C. elegans* zygotes. The positioning of intracellular structures and organelles is important for proper functioning and division of cells. While various forces inside cells are involved, organelle motion strongly depends on the drag generated by cell cytoplasm. Despite some recent advances in force measurement inside cells, little has been known about the cytoplasmic drag on different intracellular structures due to their geometric complexity (witness the centrosomal MT array) and the confinement of the cell. That said, recent advances in modeling and computation have allowed estimates of the influence of cellular confinement and associated MT arrays upon drag coefficients of nuclei.

De Simone et al use quantitative microscopy, genetic perturbations, and a simple organizing force-balance model to assess and measure the drag and force couplings on the centrosomal MT arrays and the cell nuclei during first cell division in *C. elegans* embryos. In particular they establish that the female pronucleus and male pronuclear complex (which includes the centrosomal arrays) have velocities in register as they approach each other. They use this observation to estimate the ratio of their drag coefficients, and find this is in close agreement with the new theoretical estimates of Nazockdast et al (2017) that take into account cellular confinement and MT array drag. The authors then used this data, together with genetic perturbations that selectively remove various sources of force and coupling, to estimate forces on the pronuclei as they migrate towards the cell center. These forces they ascribe to cortically bound dynein force generators, and to an additional but sub-dominant "centering force" (e.g. cytoplasmic dyneins pulling upon centrosomal MTs).

The paper is well-organized, concise and well-written, the data is of high quality, and the results are important and interesting. I am happy to recommend publication in Nature Communications.

However, a few issues should be addressed before publication and can improve the study:

> We thank the Reviewer for her/his very positive assessment of our work. We have addressed the points and questions raised by this Reviewer as detailed below.

1: The authors mentioned that cortical forces are generated by the GOA-1/GPA-16, GPR-1/2, and LIN-5 complexes, and conjecture that the centering forces are caused by cytoplasmic dynein. It is worth mentioning other sources of pulling forces as described in Gusnowski and Srayko, JCB 2011, which can act as centering forces but are independent of GOA and GPR pathways.

> We thank the Reviewer for this suggestion. We now mention these GOA-1/GPA-16 and GPR-1/2-independent centering forces in the Discussion of the revised manuscript (p. 14).

2: There are various mutations that can significantly change the size of the pronuclei in *C. elegans* embryos. It would be interesting to measure the speed of pronuclei migration in these mutants and compare it to the prediction of the model knowing the size of the pronuclei and the drag.

> Although interesting in principle, this suggestion comes with some complications. Pronuclei are indeed significantly smaller than wild-type in several conditions with a perturbed nuclear membrane, such as in *ran-1(RNAi)* and *lmn-1(RNAi)* embryos. However, a smaller pronuclear surface can cause temporary detachment of centrosomes from the male pronucleus (Meyerzon et al., Dev Biol, 2009, PMID:

19162001), complicating the determination of the drag coefficient. Due notably to these consideration, we decided not to launch these experiments. Nevertheless, the importance of nuclear size on the drag coefficient of the male-asters complex is mentioned in the Discussion (p. 13-14).

3: Following the theoretical study in (14) by Nazockdast and coworkers, there are two major drag coefficients associated with the motion of pronuclei: translational drag and rotational drag. While the authors well characterized the translational drag, they can perform a similar analysis to estimate the rotational drag as well. This can be an interesting addition to this paper, or as a future continuation of this study. (I see that the paper by Nazockdast et al is now published in MBOC. Hence, the authors should update that citation).

> This is a very interesting suggestion, which would indeed be attractive for a follow-up study. Analysis of rotational drag will require comparing the torsional momentum of the forces exerted on the male-asters complex and female pronucleus. This comparison will require estimating the rotational velocities of the male-aster complex and female pronucleus. While the former can be calculated by measuring the relative position of centrosomes on the male pronucleus, the latter will require marking in some way the membrane of the female pronucleus to track its rotations. We also thank the Reviewer for spotting the outdated citation, which we have updated (p. 4, 8, 11, 13 and 33).

Appendix – Minimal mathematical model of centering forces

The analysis of embryos depleted of cortical and nuclear motors showed that centering forces exhibit sigmoidal dynamics, characterized by an initial acceleration, followed by a deceleration when the asters reach the cell center (see Main Text – revised Fig. 2, 4g). Theoretical models and experiments led to the proposal that centration is driven by dynein motors that exert forces along microtubules while being bound to vesicles or another stable network in the cytoplasm (Kimura K. and Onami S., PNAS, 2005, PMID: 15866166; Kimura K. and Kimura K., PNAS, 2011, PMID: 21173218). In this model, since dynein motors in the cytoplasm can bind at any position along the length of microtubules, the total force exerted on each microtubule is proportional to its length. Therefore, more force is exerted on longer microtubules and the microtubule aster will be pulled toward the center of the cell, where forces balance each other and movement stops. This model predicts that a microtubule aster decelerates when reaching the cell center, thus providing a plausible explanation for the deceleration phase of centrosome centration in *C. elegans*, but not for the acceleration observed initially.

What could be responsible for the observed acceleration? One possibility is that such acceleration could be driven by the progressive growth of microtubules toward the anterior of the cell; as a result, these microtubules can bind to increasing number of motors (Kimura K. and Onami S., PNAS, 2005, PMID: 15866166). However, microtubule tip tracking shows that microtubule within asters grow at a speed of $\sim 0.7\text{-}1 \mu\text{m s}^{-1}$, so that they can span the whole embryo length in $\sim 50\text{-}70 \text{ s}$, whereas the acceleration phase lasts for several hundreds of seconds after the start of nucleation of microtubule asters (see Fig. 1c, 2d) (Srayko M et al., Dev Cell, 2005, PMID: 16054029). Thus, microtubule growth alone is unlikely to be the cause of centration acceleration.

A second possibility is that centrosome acceleration is driven by the size increase of centrosomes, which accumulate increasing amounts of the microtubule nucleator γ -tubulin (Decker M. et al., Curr Bio, 2011, PMID: 21802300; Zwicker D. et al, PNAS, 2014, PMID: 24979791). As a result, the number of centrosomal microtubules could increase over time, leading to an increase of total force and thus centrosome acceleration. We investigated centration in this second scenario by developing a 1D minimal mathematical model based on length-dependent forces that includes an increase in the number of microtubules. In the model, each aster is composed of N microtubules, which can be either directed toward the anterior or the posterior side. Since centrosome size increases exponentially over time, we assume that the number of microtubules increases exponentially over time as well (rate α). Microtubules have a fixed length proportional to the distance from the centrosome pair to the cortical side towards which microtubules are directed (proportionality factor s). Force is exerted proportionally to the concentration of cytoplasmic motors n and microtubule length (proportionality factor f). Thus, the total force exerted from posterior (F_+) and anterior (F_-) microtubules reads

$$F_+ = \frac{nf}{2} N s(L - xL)$$

$$F_- = -\frac{nf}{2} N sxL$$

where x is the position of the centrosome pair in normalized coordinates and L is embryo length. The number of microtubules per centrosome increases over time as

$$N(t) = N_0 e^{\alpha t}$$

Thus, the total force reads

$$F = F_+ + F_- = nfL N_0 e^{\alpha t} \left(\frac{1}{2} - x \right)$$

Centrosomes move at a velocity $v = \frac{F}{\gamma_A}$ in response to the total force F , where γ_A is the drag coefficient of the aster pair. To set how the drag coefficient of the aster pair scales with microtubule number, we analyzed the predictions of Nazockdats et al. (MBoC, 2017, PMID: 28331070). In that work, computational modeling predicts that the drag force of the male-asters complex scales with the number of microtubules as $\gamma_{MAC} = \gamma_M + rN^\beta$, where $\beta \approx 0.7$ and γ_M is the drag coefficient of the male pronucleus when there are no microtubules (fit performed on the data of Fig. 2A in Nazockdats et al. (MBoC, 2017, PMID: 28331070)). We assume that when the male pronucleus is detached as in *zyg-12(ct350) goa-1/gpa-16(RNAi)* embryos, the drag coefficient of the microtubule aster pair scales with microtubule number. Since in this condition the male pronucleus is detached and thus does not contribute to the drag, we assume the drag coefficient of the aster pair to be $\gamma_C = rN^\beta$, where the drag coefficient of the two centrosomes is considered negligible.

Therefore, centrosome centration position reads as the sigmoid

$$x(t) = x_{eq} + (x_0 - x_{eq}) e^{-ce^{dt}} \quad (\text{Eq. 4})$$

where $c = \frac{nfLN_0^{1-\beta}}{r\alpha(1-\beta)}$ and $d = \alpha(1 - \beta)$. Overall, we conclude that microtubule length-dependent forces, together with an exponential increase in the number of microtubules, can explain the sigmoidal dynamics of centrosome centration.

Reviewer #1 (Remarks to the Author):

The authors have addressed my concerns in a satisfactory manner. I now support the publication of the manuscript.

The manuscript contains valuable quantitative data for researchers in the field. I thus recommend the authors to publish the quantitative data, in particular v_{MAC} and v_F against t of *goa-1/gpa-16*(RNAi) embryos that they used for the model selection analysis, as supplementary materials (e.g. Excel sheets).

Reviewer #2 (Remarks to the Author):

The authors have responded well to reviewer concerns. This work is an interesting step forwards in understand the spatial organization of large embryo cells. I dont think its the final word on aster movement, but I do think it should be published in its current form

Point-by-point response to the reviewers' comments

We thank the reviewers for their appreciation and positive assessment of our work.

Reviewer #1 (Remarks to the Author):

The manuscript contains valuable quantitative data for researchers in the field. I thus recommend the authors to publish the quantitative data, in particular v_{MAC} and v_F against t of *goa-1/gpa-16(RNAi)* embryos that they used for the model selection analysis, as supplementary materials (e.g. Excel sheets).

> We thank the Reviewer for her/his suggestion. Following this recommendation, we added to the revised manuscript an Excel File (Supplementary Data 1) in which we report the positions of pronuclei and centrosomes as a function of time for all embryos analyzed in this work. Positions provide more information than mere velocities, which can be easily calculated from positions.